A survey on sleep assessment methods

Ibáñez Vanessa vanessa.ibanez@ucv.es 1
Silva Josep 2
Cauli Omar 3
1 Facultad de Enfermería, Universidad Católica de Valencia “San Vicente Mártir” , Valencia , Spain
2 Departamento de Sistemas Informáticos y Computación, Universitat Politècnica de València , Valencia , Spain
3 Departamento de Enfermería; Universidad de Valencia , Valencia , Spain
Abdullah Jafri
Electronic publication date: 2018 May 25
Publication date: 2018
Volume: 6
Electronic Location ID: e4849
Received 2018 Feb 17; Accepted 2018 May 7
Copyright: ©2018 Ibáñez et al.
Copyright year: 2018
Copyright holder: Ibáñez et al.
License: This is an open access article distributed under the terms of the Creative Commons Attribution License, which permits unrestricted use, distribution, reproduction and adaptation in any medium and for any purpose provided that it is properly attributed. For attribution, the original author(s), title, publication source (PeerJ) and either DOI or URL of the article must be cited.
License URL: https://creativecommons.org/licenses/by/4.0/

Keywords: Sleep, Sleep assessment, Sleep disorders, Sleep assessment methods

Funding: The authors received no funding for this work.

==============================
Purpose

A literature review is presented that aims to summarize and compare current methods to evaluate sleep.

Methods

Current sleep assessment methods have been classified according to different criteria; e.g., objective (polysomnography, actigraphy…) vs. subjective (sleep questionnaires, diaries…), contact vs. contactless devices, and need for medical assistance vs. self-assessment. A comparison of validation studies is carried out for each method, identifying their sensitivity and specificity reported in the literature. Finally, the state of the market has also been reviewed with respect to customers’ opinions about current sleep apps.

Results

A taxonomy that classifies the sleep detection methods. A description of each method that includes the tendencies of their underlying technologies analyzed in accordance with the literature. A comparison in terms of precision of existing validation studies and reports.

Discussion

In order of accuracy, sleep detection methods may be arranged as follows:

Questionnaire < Sleep diary < Contactless devices < Contact devices < Polysomnography

A literature review suggests that current subjective methods present a sensitivity between 73% and 97.7%, while their specificity ranges in the interval 50%–96%. Objective methods such as actigraphy present a sensibility higher than 90%. However, their specificity is low compared to their sensitivity, being one of the limitations of such technology. Moreover, there are other factors, such as the patient’s perception of her or his sleep, that can be provided only by subjective methods. Therefore, sleep detection methods should be combined to produce a synergy between objective and subjective methods. The review of the market indicates the most valued sleep apps, but it also identifies problems and gaps, e.g., many hardware devices have not been validated and (especially software apps) should be studied before their clinical use.

Introduction

Sleep is fundamental to health. Sleep disorders can often be a symptom of a disease; or also may be an indicator of a future disease such as depression. For those reasons, sleep assessment is an essential component of any health check. As such, many health care systems stablish mechanisms to prevent sleep disorders by providing specific plans in relation to education and awareness of good sleep habits.

Over the years, many different sleep assessment methods have appeared. Specially in the last years, new methods have emerged with the appearance of new technologies such as mobile apps and novel advanced hardware sensors such as galvanic skin response measurers. In this survey, we review the current methods for the detection of sleep. From simple methods that only distinguish between awake or asleep states to complex methods able to distinguish all the sleep stages; from subjective methods such as sleep questionnaires and sleep diaries to objective methods such as polysomnography.

The main motivation of this survey is to produce a comprehensive and unbiased literature review from which we can extract a complete classification of sleep assessment methods (including new technologies such as mobile apps). There have in the past been different reviews of sleep assessment methods, but most of them are outdated (see, e.g., Lomeli et al., 2008; Kelly, Strecker & Bianchi, 2012; Winter, 2014), or they are partial, or only focus on a specific subset of methods (e.g., sleep questionnaires: Silva et al., 2011; Firat et al., 2012; El-Sayed, 2012; Pataka et al., 2014; Singh & Mims, 2015; Chai-Coetzer et al., 2015, mobile apps: Lee & Finkelstein, 2015; Ong & Gillespie, 2016, or contact sleep detection methods: Kolla, Mansukhani & Mansukhani, 2016; Maslakovic, 2017; Green, 2017, etc.).

The survey was written to appeal to a range of people, who would have a broad spectrum of interests. It covers all sleep detection methods and, for each method, it also provides a table with the most used market products. Hence, because the survey tackles different technical areas, all technical terms have been conveniently introduced and explained. In all cases, explanations are supported and complemented with adequate references. Of course, not all sleep detection methods have the same precision; in fact, some of them are completely subjective. Therefore, the comparison of methods deserves a critical view on validation. Thus, we also report on the reliability and validity of the methods analyzing previous comparisons and validation studies.

Survey methodology

The literature review begins with a planning phase. This phase formulates research questions and defines inclusion and exclusion criteria. This phase is followed by search and screening of primary studies.

Research questions

We formulated two research questions to identify the current state of the art in sleep assessment methods:

• What methods for sleep assessment have been developed?

This research question aims to provide an overview of the sleep assessment methods, with special emphasis on those that have been developed over the last 10 years.

• What are the main characteristics of each sleep assessment method?

This question complements the previous one, giving a deeper understanding of the sleep assessment methods.

Search process

The purpose of a literature review is to conduct a review of relevant studies to assess the body of knowledge that exists to support addressing the research questions. This process is rigorous and unbiased, and it involves a wide coverage of sources, such as online databases, journals, and conferences. The search string created to retrieve information from the electronic resources and databases is the following:

(assessment OR evaluation OR detection)	
AND (method OR tool OR environment OR system)	
AND (sleep)	

This search string was designed after an analysis of the keywords from the relevant literature, which was found from several general searches in the resources outlined above.

With the search terms defined, we started the process of identifying relevant literature in the following electronic databases: PubMed, LILACS, TOXNET, SCOPUS, ScienceDirect, and Google Scholar. Initially, we sought potential primary studies in the databases. In PubMed Health, the search string produced 1,784 results. Therefore, we had to filter the results by refining the search string for that database:

“sleep detection”[Title/Abstract] OR “sleep assessment”[Title/Abstract]	

As a result of the search process, 318 studies were identified. Excluding unavailable and duplicated results, we obtained 212 studies.

Inclusion and exclusion criteria

To address the research questions, the following inclusion and exclusion criteria were defined:

• IC1: Those papers that discussed sleep assessment methods were included.

• IC2: Those papers that described the characteristics of a sleep assessment method were included.

• EC1: Those papers that did not describe a sleep assessment method were excluded.

Studies selection

Initially, we performed screening on the titles and abstracts to decide whether to include or exclude each study. As a result, from the six sources that we searched, a total of 114 studies were selected and 98 were excluded. We read in detail the full text of each primary study included in the preliminary selection to decide whether to include or exclude the study. The primary studies included in the final selection correspond to the relevant papers that meet the research questions set out in this study. The QUOROM flow chart of the reviewing process is depicted in Fig. 1.

Figure 1 QUOROM flow chart of the reviewing process.

Solid arrows represent the QUOROM flow. Dashed arrows represent the decomposition of a box into several sub-boxes.

Data extraction

With the final set of primary studies decided upon, the data extraction activity was carried out on included papers. For each paper, we identified the kind of article (review, opinion, study, tool description …) and the sleep assessment methods it described. We grouped the data by sleep assessment methods and identified a total of five categories where all sleep assessment methods can be classified (see ‘Classification of Sleep Detection Methods’). For each method, a single document was produced, grouping the data coming from all papers related to the tool. This produced summaries and charts that helped us to study and classify the methods.

Structure of the survey

The rest of the paper has been structured as follows: in ‘Classification of Sleep Detection Methods’, a classification of sleep detection methods is proposed. Then, in ‘Medical Assistance Methods’, those methods that need medical assistance are explained. Similarly, those methods that do not necessarily need medical assistance (self-assessment methods) are explained in ‘Self-Assessment Methods’. In ‘A Critical Discussion About Accuracy and Validation’, we discuss the accuracy and validity of the methods presented. We also discuss the usefulness of some of the methods, and we comment on future developments. Finally, in ‘Conclusions’ we provide a concluding summary.

Classification of Sleep Detection Methods

Essentially, a sleep detection method is a function that classifies the sleep state of a patient. Most sleep detection methods such as wrist actigraphy or mobile apps consider a binary function, where the state can be classified as Awake/Sleep. More sophisticated methods consider a ternary function: Awake/NREM/REM. And, finally, the most advanced methods, such as polysomnography—often used as the gold standard—consider a quinquenary function: Awake/N1/N2/N3/REM. Hence, any method can produce a two-dimensional chart where the X-axis is Time, and the Y-axis is the State of the Patient. In the particular case of polysomnography, the Y-axis has five possible values; thus, it can determine the sleep stage of the patient at any time, and study the transitions occurring between the states. Of course, a sleep study such as a polysomnography often produces much more complementary information that can be used, e.g., to diagnose sleep diseases. Among the information reported by a polysomnography we find oxygen saturations, limb movements, apneas, respiratory events by body position, etc. The interested reader is referred to Robertson, Marshall & Carno (2014), Pandi-Perumal, Spence & BaHammam (2014) and Armon et al. (2016) for information about sleep study reports and their interpretation and usage.

The information that is common to the majority of sleep detection methods is the one that refers to a binary state classification (i.e., Awake/Sleep), because this is achieved by the basic methods, and subsumed by the advanced methods. Table 1 defines the basic parameters that can be collected by a binary state classification method. In grey, we show the primary data that should be collected by the sleep detection device and, in white, we show the most important parameters that can be derived from the primary data.

Table 1 Definition of basic sleep detection parameters.

This table summarizes the main parameters of a sleep study. The top of the table (light blue) lists the fundamental parameters. Those parameters that can be derived from the primitive variables are listed in the dark blue rows. Each of them includes its associated formula.

Sleep measure	Definition	Formula	
Fundamental parameters	
Initial In Bed Time (IIB)	Time when patient goes to bed initially	–	
Final Out Bed Time (FOB)	Time when patient leaves the bed definitely	–	
Time Out of Bed (TOB)	Total time out of bed between IIB and FOB	–	
Lights Out Time (LT)	Time of lights out	–	
Lights On Time (LN)	Time of lights on	–	
Sleep Onset (SO)	Time when first sleep starts	–	
Final Sleep (FS)	Time when last sleep finishes		
Sleep Latency (SL)	Time taken to fall sleep (at any time)	–	
Sleep Period (SP)	Time spent sleeping between two awakenings/SO	–	
Awake Period (AWP)	Time spent awake between two sleep periods	(awakening=wake period >10 s)	
Arouse Period (ARP)	Time spent awake between two sleep periods	(arousal=wake period <10 s)	
Derived parameters	
In Bed Time (IBT)	Total time in bed	IBT = FOB-IIB-TOB	
Total Recording Time (TRT)	Time between lights out and lights on	TRT = LN-LT	
Initial Sleep Latency (ISL)	Time taken to fall sleep the first time	ISL = SO-LT	
Total Sleep Time (TST)	Amount of time the patient sleeps during TRT	TST=∑i=1i=N#sleepperiodsSPi	
Sleep Interval (SI)	Time between the first sleep and the last sleep	SI = FS-SO	
Wake After Sleep Onset (WASO)	Wake time between IIB and FOB	WASO = SI-TST	
Total Wake Time (TWT)	All wake time throughout TRT	TWT = ISL +WASO	
Mean Sleep Latency (MSL)	Arithmetic average of sleep latencies	MSL=∑i=1i=N#sleeplatenciesSLi∕N	
Sleep Efficiency (SE)	Percentage of sleep of the total time in bed	SE = (TST/TRT) ×100	
Mean Awakening Length (MAL)	Arithmetic average of awake periods	MAL=∑i=1i=N#awakeperiodsAWPi∕N	
Awakening Index (AWI)	Number of awakenings per unit of time	AWI = #AWP/TST	
Arousal Index (ARI)	Number of arousals per unit of time	ARI = #ARP/TST	

These parameters are particularly useful to determine the kind of sleep of patients, and each single parameter is relevant for a different sleep disorder or disease. For instance, the sleep onset, sleep latency, and total sleep time are essential to diagnose patients with insomnia. Similarly, an excess in the awakening and arousal indices suggests increased sleep fragmentation. In addition to the number of sleep states that they are able to detect, a sleep detection method can be classified according to other functional and operational characteristics, such as their underlying technology, which in turn directly affects their precision and validity.

In Fig. 2, we present a taxonomy of sleep detection methods. They all can be classified into two main groups according to whether they need medical assistance (Medical Assistance) or not (Self-Assessment). In this respect, there are methods that have been classified as not requiring medical assistance, such as Questionnaires and Sleep Diaries, even though their interpretation should be normally done by a professional. However, in the current state of the art there are many systems such as mobile apps that provide custom sleep questionnaires and produce reports without medical assistance. Hence, they are classified as Self-Assessment. They both deserve a deep discussion and will be explained separately in ‘Medical Assistance Methods’ and ‘Self-Assessment Methods’, respectively.

Figure 2 Taxonomy of sleep detection methods.

Grey boxes represent categories. White boxes represent sleep assessment methods or technology used to assess sleep.

Self-Assessment methods include subjective methods such as questionnaires and sleep diaries (the figure lists some instances), and objective methods based on hardware sensors, which in turn can be classified as Contact devices or Contactless devices, depending on whether they need to be in contact with the patient’s body during sleep. Those devices that are based on the echo produced by signals can be further classified into Sonar, Radar, and Lidar devices. All of them will be explained in a dedicated section.

Medical Assistance Methods

There are different studies that can be performed in a sleep laboratory. All of them have one significant advantage and one significant disadvantage that differentiate them from the home detection methods. The obvious advantage is that these methods can use advanced technology such as electroencephalograms, electrocardiograms, etc. that cannot be used at home. The advantage of these methods is that they can be extremely precise, and can be discrete (e.g., are able to distinguish between sleep phases). For this reason, these methods have been often used as the gold standard for sleep evaluation (see, e.g., Silva et al., 2011; Firat et al., 2012; El-Sayed, 2012; Luo et al., 2014; Chai-Coetzer et al., 2015; Silva et al., 2016). Of course, the use of this exclusive technology comes with a cost: these methods are expensive, time-consuming, require professional assistance and, often, they can only be done for a reduced period of time (e.g., one or two days). But, additionally, there is another important functional disadvantage: the assessment made by these methods is done in a context that is not the usual sleep context of the patient (i.e., a sleep clinic or a hospital) and, thus, a normal sleep situation is not measured.

Polysomnogram (PSG)

The term polysomnogram comes from the Greek root poly (many), the Latin noun somnus (sleep), and the Greek verb noun gramma (drawing or diagram). A PSG (Robertson, Marshall & Carno, 2014; Pandi-Perumal, Spence & BaHammam, 2014; Armon et al., 2016) is a medical procedure composed of several concurrent but independent tests that monitor different body functions during sleep and that are recorded for their later study using different channels. An exhaustive list of tests and information gathered in a modern PSG follows:

∘ Electroencephalogram (EEG)—measures and records the brainwave activity to identify sleep stages and detect seizure activity.

∘ Electrooculogram (EOG)—records eye movements. These movements are important for identifying the different sleep stages, especially the REM stage.

∘ Electromyogram (EMG)—records muscle activity (e.g., teeth grinding and face twitches; but also, limb movements using surface EMG monitoring of limb muscles, periodic or other). Chin EMG is necessary to differentiate REM from wakefulness, limb EMG can identify periodic limb movements during sleep (PLMS).

∘ Electrocardiogram (EKG)—records the heart rate and rhythm.

∘ Pulse oximetry—monitors the oxygen saturation (SO2).

∘ Respiratory monitor—measures the respiratory effort (thoracic and abdominal). It can be of several types, including impedance, inductance, strain gauges, etc.

∘ Capnography—measures and graphically displays the inhaled and exhaled CO2 concentrations at the airway opening.

∘ Transcutaneous monitors—measure the diffusion of O2 and CO2 through the skin.

∘ Microphone—continuously records the snoring volume and kind.

∘ Video camera—continuously records video. It is useful to identify the body motion and position.

∘ Thermometer—records the core body temperature and its changes.

∘ Light intensity tolerance test—determines the influence of light intensity on sleep.

∘ Nocturnal penile tumescence test—is used to identify physiological erectile dysfunctions.

∘ Esophageal tests—includes pressure manometry, to measure pleural pressure; esophageal manometry to assess peristalsis, and esophageal pH monitoring (acidity test).

∘ Nasal and oral airflow sensor—records the airflow and the breathing rate.

∘ Gastroesophageal monitor—is used to detect Gastroesophageal Reflux Disease (GERD).

∘ Blood pressure monitor—measures the blood pressure and its changes.

Depending on the particular sleep study that needs to be performed, only some specific tests from the above list are generally selected—and they are also parameterized for each specific case. For instance, the EEG is usually comprised of 10–16 electrodes, but in patients with epilepsy, often 20 electrodes are used. Similarly, to assess bruxism, the EMG electrodes can be placed over the masseter muscle, but to assess other sleep disorders, the EMG electrodes are placed in other muscle groups. For example, the intercostal EMG is used to measure the effort during respiration.

Nowadays, the PSG is the most advanced tool for the diagnosis of many sleep disorders. According to Pandi-Perumal, Spence & BaHammam (2014) and Armon et al. (2016), the main disorders that a PSG can evaluate are those in Fig. 3 (they are classified following the International Classification of Sleep Disorders (Sateia, 2014)), being especially frequent: (i) sleep apnea or another sleep-related breathing disorder, (ii) periodic limb movement disorder, (iii) narcolepsy, (iv) REM sleep behavior disorder, (v) unusual behaviors during sleep, and (vi) unexplained chronic insomnia.

Figure 3 Classification of the main disorders evaluated with a polysomnography.

The main disorders evaluated with a polysomnogram are structured with a three-levels taxonomy that follows the International Classification of Sleep Disorders.

Multiple sleep latency test (MSLT)

This sleep study (Carskadon, 1986; Sullivan & Kushida, 2008) is a test to identify excessive daytime sleepiness (i.e., feeling sleepy in a situation where one should be awake and alert, e.g., driving a truck) and determines how long it takes the patient to fall asleep. It also identifies the phases of the sleep (e.g., how quickly and how often the patient enters REM sleep). MSLT is the standard test to diagnose idiopathic hypersomnia and narcolepsy, and it measures how quickly a patient falls asleep during the day in a quiet environment. An MSLT often starts the morning following a PSG and it lasts one complete day. The patient tries to sleep in five scheduled naps separated by two-hour breaks. For this reason, this test is often called a “nap study”.

Each nap trial takes place in a quiet bedroom. The patient is connected with sensors to a device that can detect sleep stages. The standard procedure often includes an EEG, EOG, EMG, and EKG (Carskadon, 1986). The equipment is composed of different electrodes and monitors:

∘ Wires with small cup electrodes attached to the scalp with a conductive paste to measure brain activity (EEG). This detects in what stage of sleep is the patient.

∘ Wire electrodes that are taped to the face near the eyes (EOG) and chin to show muscle activity (EMG).

∘ Two elastic belts around the chest and stomach to measure breathing effort.

∘ A nasal cannula and small heat monitor to measure all breathing activity.

∘ A wire electrode on each leg to measure body movement/muscle activity.

∘ A monitor taped to a finger to detect oxygen levels.

∘ Two to three lead EKG monitors to show heart rate and rhythm.

∘ A small microphone applied to the throat to detect snoring.

Hence, the MSLT can identify exactly when the patient falls asleep, and whether or not they entered REM sleep. If the patient falls asleep, they are awakened after 15 min. The nap trial also ends if the patient does not fall asleep within 20 min. Patients with narcolepsy often have two or more REM periods during the MSLT. People with idiopathic hypersomnia fall asleep easily but do not reach REM sleep during the nap trial.

Maintenance of wakefulness test (MWT)

This test (Banks et al., 2004; Meira et al., 2017) is performed over a whole day. Contrary to a PSG, this test is made while the patient is awake. Essentially, it challenges patients to attempt to stay awake during periodic tests. Therefore, an MWT may be helpful in the management of sleepy patients, particularly for driving purposes. It measures how alert a patient is during the day and it determines whether a patient is able to stay awake for a period of time in a quiet and relaxing environment. During the test, there are four to five periods of around 40 min each, spaced apart by 2 h, where the patient is asked to stay relaxed in a quiet, faintly-lit bedroom. The first trial often begins 1.5 to 3 h after the patient’s normal wake-up time. The patient eats breakfast one hour prior to the first relaxing period and they have lunch after the second period. Between the periods, the patient can read the newspaper, watch TV, have a meal, or move freely inside the building, but they cannot go outside because daylight is a factor that must be eliminated during the test.

During the relaxing periods, patients are connected to a set of leads that monitor (i) heart activity with two to three ECG leads, (ii) brain activity with 4 EEG leads, (iii) chin muscle activity with three leads, and (iv) left and right eye movements. If the patient falls asleep for 90 s at any time during the relaxing period, the test is terminated. All data collected are analyzed by a sleep specialist to determine the patient’s level of sleepiness during the day.

CPAP titration test (CTT)

A CTT (Lopez-Campos et al., 2007) is a type of sleep study that is used to calibrate continuous positive airway pressure (CPAP) and bi-level positive airway pressure (BIPAP) therapies. CPAP/BIPAP are the common treatments in some sleep-related respiratory disorders (see Fig. 3) such as central sleep apnea (BIPAP) and obstructive sleep apnea (CPAP), which eliminate breathing pauses during sleep. Before starting these treatments, a CTT is needed.

The objective of a CTT is to determine the amount of air pressure needed to prevent the upper airway from becoming blocked. This is studied during the sleep of the patient with a nasal mask that periodically changes the air pressure, and different sensors that monitor the sleep in a similar way to a PSG (i.e., they record oxygen levels, breathing, heart rate, brain waves, and leg and arm movements).

Home sleep test (HST)

The HST (Cruz, Littner & Zeidler, 2014; Kapoor & Greenough, 2015) is a kind of limited PSG that is made at home (i.e., portable equipment is transported to the patient’s home). The number of channels used is often reduced to three: airflow, respiratory effort, and oximetry. It provides an indication only for high suspicion of obstructive sleep apnea—not other sleep disorders—and it has the obvious advantage that the context in which the sleep is evaluated is the normal one. The main disadvantages are that it cannot determine sleep stage, hypopneas, or arousals; and no one is present to replace leads.

Self-Assessment Methods

Sleep questionnaires

The preliminary evaluation of sleep in primary care is often completed with a sleep questionnaire (also known as a sleep scale). Sleep questionnaires are a very inexpensive and rapid test, and for these reasons, they are ideal for the first diagnostic test. Moreover, they summarize in a quantitative way the (subjective) perception of the patient about his or her own quality of sleep. Precisely because they are mostly subjective, sleep questionnaires can be influenced by the same sources of bias and inaccuracy as any other such reports. However, their subjectivity does not necessarily render questionnaires inaccurate, as it has been demonstrated by several validation studies (see Silva et al., 2011; El-Sayed, 2012; Firat et al., 2012; Luo et al., 2014; Pataka et al., 2014; Chai-Coetzer et al., 2015).

In general, filling in a sleep questionnaire does not require the assistance of sanitary professionals. They can be self-administered at any moment, even at home. For instance, the Google play’s Sleep Apnea Screener is a mobile app that automatically provides a report after completing a questionnaire. Therefore, sleep questionnaires can be used by people (e.g., with sleep apnea) as a sleep control that can alert them about the need for a proper diagnosis provided by specialists.

Table 2 shows (in chronological order of appearance) the most extended sleep questionnaires used along the last 30 years. For a long time, we have been collecting all of them (some of them are not available online), and we have created a public repository where they all can be downloaded: http://users.dsic.upv.es/ jsilva/Sleep/.

Table 2 Questionnaires for the detection of sleep disorders.

Each row represents a sleep questionnaire, and includes its acronym, its structure (number of items and scale used), and a reference to the article where it was proposed.

Sleep questionnaire	Structure	Period	Objectivity	
MSQ	Mini Sleep Questionnaire (Zoomer et al., 1985)	10 items (7 point scale)	Recently	0	
PSQI	Pittsburgh Sleep Quality Index (Buysse et al., 1989)	9 items (4 point scale)	1 month	0	
ESS	Epworth Sleepiness Scale (Johns, 1991)	8 items (4 point scale)	Recently	0	
ISI	Insomnia Severity Index (Morin, 1993)	7 items (5 point scale)	Recently	0	
SDQ	Sleep Disorders Questionnaire (Douglass et al., 1994)	175 items (5 point scale)	Recently	1	
SACS	Sleep apnea clinical score (Flemons et al., 1994)	4 items (100 point scale)	Recently	4	
FOSQ	Functional Outcomes of Sleep Questionnaire (Weaver et al., 1997)	30 items (4–5 point scale)	Recently	0	
SAQLI	Calgary Sleep Apnea Quality of Life Index (Flemons & Reimer, 1998)	35 items (7 point scale)	1 month	0	
OSQ	Oviedo Sleep Questionnaire (Bobes et al., 1998)	15 items (4–7 point scale)	1 month	0	
BQ	Berlin Questionnaire (Netzer et al., 1999)	10 items (2–5 point scale)	Recently	2	
ASQ	Athens Sleep Questionnaire (Soldatos, Dikeos & Paparrigopoulos, 2000)	8 items (4 point scale)	1 month	0	
SEMSA	Self-efficacy in Sleep Apnea (Weaver et al., 2003)	26 items (4 point scale)	Recently/Future	0	
SQ	STOP Questionnaire (Chung et al., 2008)	4 items (2 point scale)	Recently	2	
SBQ	STOP-BANG Questionnaire (Pallesen et al., 2008)	8 items (2 point scale)	Recently	3	
BIS	Bergen Insomnia Scale (Chasens, Ratcliffe & Weaver, 2009)	6 items (8 point scale)	1 month	0	
FOSQ-10	Functional Outcomes of Sleep Questionnaire—10 (Takegami et al., 2009)	10 items (4 point scale)	Recently	0	
SFV	Simple Four Variables (Chai-Coetzer et al., 2011)	4 items (2–6 point scale)	Recently	3	
OSA50	Obesity, Snoring, Apneas, aged over 50 (Chai-Coetzer et al., 2011)	4 items (3–4 point scale)	Recently	4	

For each questionnaire, the table shows:

• Its structure: number of questions/items and the scale used for the answers.

• The period of time that the questionnaire evaluates: if it is unspecific or unspecified it uses “Recently”, if one or more questions refer to future or hypothetical situations it uses “Future” (e.g., “If I use CPAP I will feel better”, “I would use CPAP, even if I had to pay for some of the cost”, etc.).

• The percentage of objective questions in the questionnaire: a question that is (partially) subjective or that depends on memory is considered subjective. Only questions that are totally objective are considered objective (e.g., “have you taken drugs to sleep?”, “how much do you weight?”, etc.). The level of objectivity is indicated with a 1–4 scale, where 0 means close to 0%, 1 means close to 25%, 2 means close to 50%, 3 means close to 75%, and 5 means close to 100%.

It is important to note that, although the goal of some questionnaires (e.g., PSQI) is to evaluate sleep quality (such as PSG, and actigraphy), others assess concepts distinct from sleep quality. For example, FOSQ measures the concept of sleepiness, which may or may not be related to sleep quality. Treating objective and subjective measures related to some aspect of sleep as evaluating sleep quality would ignore the fundamental concept on which they were developed and the principle that you select a measure based on the concept you are measuring for alignment and accuracy. Therefore, some questionnaires are fundamentally incomparable, and the selection of one questionnaire should be based on the purpose of each specific questionnaire. Table 3 summarizes the objective of each questionnaire.

Table 3 Purposes of sleep questionnaires.

Each row represents a sleep questionnaire, and it indicates what does this questionnaire intend to measure.

Sleep questionnaire	Acronym	Measures	
Mini Sleep Questionnaire	MSQ	Insomnia and hypersomnia	
Pittsburgh Sleep Quality Index	PSQI	Sleep quality and patterns of sleep in adults	
Epworth Sleepiness Scale	ESS	Level of daytime sleepiness. Average sleep propensity in daily life	
Insomnia Severity Index	ISI	Nature, severity, and impact of insomnia. Treatment response in adults	
Sleep Disorders Questionnaire	SDQ	Sleep disturbance and usual sleep habits during the past month only	
Sleep apnea clinical score	SACS	Sleep apnea	
Functional Outcomes of Sleep Questionnaire	FOSQ	Impact of excessive sleepiness on daily life	
Calgary Sleep Apnea Quality of Life Index	SAQLI	Quality of life associated with sleep apnea	
Oviedo Sleep Questionnaire	OSQ	Insomnia and hypersomnia in the last month	
Berlin Questionnaire	BQ	Sleep apnea	
Athens Sleep Questionnaire	ASQ	Sleep quality	
Self-Efficacy Measure for Sleep Apnea	SEMSA	Sleep apnea	
STOP Questionnaire	SQ	Sleep apnea	
STOP-BANG Questionnaire	SBQ	Sleep apnea	
Bergen Insomnia Scale	BIS	Sleep quality	
Functional Outcomes of Sleep Questionnaire—10	FOSQ-10	Impact of excessive sleepiness on daily life	
Simple Four Variables	SFV	Sleep apnea	
Obesity, Snoring, Apneas, aged over 50	OSA50	Sleep apnea	

Having such an availability of different questionnaires (as shown in Table 2, their number of questions and scales vary a lot), the natural question is: “Which sleep questionnaire should I use?” Of course, those questionnaires with less questions are easier to administer, but those questionnaires with more questions collect more information. However, the question remains for those questionnaires with the same number of questions (e.g., SACS, SQ, SFV, OSA50). This question has motivated several studies to compare their sensitivity (true positive rate) and specificity (true negative rate). Some important studies comparing sleep questionnaires for the identification of sleep apnea are summarized in Table 4. The interested reader is referred to Ibáñez, Silva & Cauli (2018) for a survey on sleep questionnaires.

Table 4 Studies that compare sleep assessment questionnaires.

Each row represents a study that compares 3–5 sleep questionnaires. For each study, the table shows the size of the sample used (amount of people that participated in the study) and which questionnaire produced the best sensitivity and specificity. The reference to each study is also included.

Questionnaires evaluated	Sample	Best sensitivity	Best specificity	Reference	
ESS vs. SQ vs. SBQ vs. SFV	4,770	SBQ (87.0%)	SFV (93.2%)	Silva et al. (2011)	
ESS vs. BQ vs. SQ vs. SBQ	234	SBQ (97.55%)	ESS (75.0%)	El-Sayed (2012)	
BQ vs. SQ vs. SBQ vs. OSA50	90	SBQ (87%)	SBQ (76.0%)	Firat et al. (2012)	
ESS vs. BQ vs. SQ vs. SBQ	212	SBQ (94.9%)	SFV (50.0%)	Luo et al. (2014)	
ESS vs. BQ vs. SQ vs. SBQ vs. SFV	1,853	SBQ (97.6%)	SFV (74.4%)	Pataka et al. (2014)	
SQ vs. SBQ vs. OSA50	543	OSA50+oximetry (73.0%)	OSA50+oximetry (96.0%)	Chai-Coetzer et al. (2015)	

Sleep diaries

Sleep diaries allow patients to self-assess their sleep. Sleep diaries have one important advantage over sleep questionnaires. While sleep questionnaires are filled in once, sleep diaries are filled in over a period of time (usually one or two weeks). This means that sleep diaries contain more information, and also that the information contained is more precise. This happens because a sleep questionnaire provides an overall perception, often ignoring the details, and it is highly dependent on the patient’s memory because they summarize information about the previous one or two weeks. Contrarily, the sleep diary collects data every day, so that good and bad days are recorded. Moreover, the sleep diary is not so dependent on memory, because they are often filled in just after waking up. We have been collecting sleep diaries from hospitals, sleep centres and different studies. In our repository, there are more than 25 sleep diaries. The most representative are shown in Table 5. We have made them publicly available at: http://users.dsic.upv.es/ jsilva/Sleep/.

The Pittsburgh Sleep Diary (Monk et al., 1994) is the oldest sleep diary in our records (although there is evidence that sleep diaries were in clinical use for decades before its 1994 publication (Weitzman et al., 1982)). After it was proposed, many other diaries have been defined by researchers, hospitals, and sleep centres. In March 2005, 25 researchers attending the Pittsburgh Assessment Conference developed an initiative to compare a collection of sleep diaries in order to extract the best from each diary studied and integrate all together, producing an improved sleep diary. As a result, they proposed the “Consensus Sleep Diary” (Carney et al., 2012) (see Table 5). The diaries included in Table 5 are classified according to the information required from the user. In particular, they include different questions about their sleep such as time used to fall asleep, the amount and kind of food in the dinner, or use of drugs, etc. Specific information about the structure and information gathered by sleep diaries can be found at Ibáñez, Silva & Cauli (2018).

Table 5 Sleep diaries for the detection of sleep problems.

Each row represents a sleep diary, and it indicates the number of questions included in the diary and the scale used to complete the answers.

Sleep diary	Number of questions	Scale	
Pittsburgh Sleep Diary (PSD)	23	6 point scale	
Consensus Sleep Diary (CSD)	20	5 point scale	
National Sleep Foundation (NSF)	15	3 point scale	
Get Self Help Sleep Diary (GSH)	14	11 point scale	
National Heart, Lung, and Blood Institute (NHLBI)	12	3–4 point scale	
NPS MedicineWise Sleep Diary (NPS)	11	3 point scale	
Loughborough Sleep Research Center (LSRC)	8	5 point scale	

There is also the existence of sleep diaries that are distributed as mobile apps. The most used and better valued sleep diaries according to Google Play are Sleep Diary Pro (423 reviews with a mark of 4.2/5), Healthy Sleep Diary (223 reviews with a mark of 3.9/5), and Sleep Diary Lite (2,263 reviews with a mark of 3.8/5). Tonetti, Mingozzi & Natale (2016) compared the use of paper and electronic sleep diaries and concluded that they are similar with respect to their diagnostic power.

Hardware devices

Contactless hardware devices to detect the sleep

Contactless methods to assess sleep use one or more of the following technologies: Microphone (in Nakano et al. (2014) there is an informative discussion about how to quantify snoring and sleep apnea severity), video camera, (infrared) thermometer, pressure strap or belt, pillow or mattress accelerometer, echo-based devices (Lee, Hong & Ryu, 2015) such as sonar, radar, or also lidar (still under development).

There exist periodic expert’s reviews (see, e.g., Langley, 2017; Green, 2017; ASA, 2017) that rank the most valued contactless devices according to the market (e.g., Amazon reviews). However, currently the most used device to assess the sleep is the smartphone. Because a smartphone contains a microphone, a camera, and an accelerometer, it can use these hardware features to monitor sleep. This has promoted the appearance of many mobile apps to assess sleep. The main contactless sleep detection apps according to the number of reviews in Google Play are shown in Table 6. Even though several experts’ reviews (see, e.g., Hacktosleep, 2016; Maslakovic, 2017) report high performance and reliability of these apps, there are still few scientific validation studies that support this claim. In contrast, several studies report that mobile apps are not yet prepared for clinical use (Kolla, Mansukhani & Mansukhani, 2016; Ong & Gillespie, 2016; Patel, Kim & Brooks, 2017; Lorenz & Williams, 2017). For instance, a recent study (Patel, Kim & Brooks, 2017) performed with 25 children (ages: 2–14) and where a smartphone recorded data simultaneously with a PSG suggested that smartphone apps may have value in increasing the user’s awareness of sleep issues but would not yet be accurate enough to be used as a clinical tool.

Table 6 Contactless sleep detection apps (prices and reviews are taken from Google Play).

Each row shows a sleep app. Rows are sorted in descending order according to the average review mark.

App name	Developer	Price	Average review	Number of reviews	
Sleep as Android Unlock	Urbandroid Team	3.99$	4.5 out of 5	23,686	
Sleep Cycle Alarm Clock	Northcube AB	0$	4.4 out of 5	47,965	
Sleep as Android	Urbandroid Team	0$	4.3 out of 5	244,840	
Sleep Better	Runtastic	0$	4.1 out of 5	108,825	
Sleep Time	Azumio Inc.	0$	4.1 out of 5	30,418	
Smart Sleep Manager	株式会社 C2	0$	4.1 out of 5	18,805	
Good Night’s Sleep Alarm	Ateam Inc.	0$	4.1 out of 5	10,022	
SleepBot	SleepBot	0$	4.0 out of 5	51,111	
Smart Alarm Clock	Nelurra Holdings LTD	0$	3.9 out of 5	27,442	
Sleep Analyzer	A1 Brains Infotech	0$	3.1 out of 5	2,461	

Contact hardware devices to detect the sleep

Contact hardware devices to assess sleep are small devices that can be attached to the wrist, chest, ankle, or head. Some of these devices use the Cartesian representation to record the activity of the body and thus they are known as actigraphs. Most actigraphs use an accelerometer to register movements. The information collected is used to analyse sleep. Even though contemporary actigraph devices are electronic, the first actigraphs were mechanical (contrary to common belief). In fact, the first actigraphs date from the 1950s (Tryon, Bellak & Hersen, 1991).

Due to the usefulness of the information collected by actigraphs, the use of actigraphy has been included in the ICSD-3 diagnostic criteria for several circadian sleep-wake rhythm disorders. Even though there is a clear continual improvement in the precision of sensors, and in the accuracy of algorithms, the use of actigraphs for clinical diagnosis should be considered when the device and algorithm used have been validated. In particular, the algorithm used to interpret the data is of major importance, because many proprietary algorithms do not pass enough quality controls, and some of them are even worse than the human inspection of the actigraphy data (see, e.g., Boyne et al., 2013).

The performance and reliability of hardware devices have been compared by experts’ reviews (see, e.g., Maslakovic, 2017) and also by validation studies (Evenson, Goto & Furberg, 2015; Kolla, Mansukhani & Mansukhani, 2016). However, it is important to note that fitness trackers and phone apps tend to underestimate sleep disruptions and overestimate total sleep times and sleep efficiency in normal subjects (Kolla, Mansukhani & Mansukhani, 2016).

A Critical Discussion About Accuracy and Validation

Each sleep detection method has its own level of reliability and precision. If we place the presented methods in order of accuracy, as reported in the literature (Boyne et al., 2013; Evenson, Goto & Furberg, 2015; Ibáñez, Silva & Cauli, 2018), we have:

Questionnaire < Sleep diary < Contactless devices < Contact devices < PSG	

It is important to note that this formulation does not pretend to sort the methods according to their usefulness. It would be erroneous to state that self-reporting is inferior (or less useful) to more objective measures. This would fail to appreciate that data based on patient perception may be valuable in understanding sleep problems.

Being both mostly subjective, there is an important difference in the way that sleep diaries and questionnaires are completed. Questionnaires are filled in once, usually before the interview with the sleep therapist, thus, not just after waking up. Consequently, (1) the patient’s memory strongly influences the quality of the information provided (he or she has to remember his or her sleep for a week or a month); and (ii) the information provided is a summary of many sleeps, thus, losing details about special days. In contrast, sleep diaries (1) are filled in every day, and (2) they are completed just after waking up. Hence, they are potentially more accurate and less influenced by the memory of the patient. Therefore, the amount of information and accuracy of sleep diaries is objectively superior to that of questionnaires. Here again, the superiority in precision of sleep diaries does not substitute the global assessment of sleep questionnaires, and the latter may be the relevant data therefore making the questionnaire more pertinent.

The accuracy of sleep questionnaires has been widely studied (see, e.g., Silva et al., 2011; Firat et al., 2012; El-Sayed, 2012; Pataka et al., 2014; Luo et al., 2014; Chai-Coetzer et al., 2015; Silva et al., 2016, and see Table 4 where studies that evaluated sleep questionnaires are compared). All these studies used the PSG as a gold standard and tried to evaluate the sensitivity/specificity of the questionnaires in identifying sleep apnea. The sensitivity reported was in the interval 73.0%–97.6%, while the specificity reported was in the interval 50%–96% (see Table 4). Most of the studies reported the STOP-BANG questionnaire as the one producing the best sensitivity. These studies are definitely useful, and provide good indicators, but our selection of a specific questionnaire must consider the specific illness and population targeted. Precisely because they target different populations the results of these studies are not always directly comparable. For instance, the study in Silva et al. (2011) was performed on highway bus drivers in Turkey and the study in Chai-Coetzer et al. (2015) only considered men, etc.

The effectiveness of sleep diaries has been evaluated in Jungquist et al. (2015) and Tonetti, Mingozzi & Natale (2016). These studies also compare paper diaries and electronic diaries using an actigraph as the gold standard. Both studies found that, statistically, paper and electronic diaries collect the same data; thus, their accuracy and reliability is similar.

The main difference between contact and contactless devices is their underlying technology. In general, contact devices are more accurate because most of the sensors used to monitor sleep are strongly dependent on their distance from the patient (the closer the better). A good example are the accelerometers, which are sensors used in both contact devices such as wrist watches, and contactless devices such as mattress or pillow clips. It is fairly evident that it is much more reliable to directly measure the movements of the body than approximating them by measuring the movements of the mattress or pillow. The same happens with sonars, for example. A phone’s microphone and speaker using ultrasounds as a sonar have an effective range of about 1 metre and its reliable distance is 0.5 m. Of course, the results are more precise as the patient is closer. Unfortunately, the movements of the sonar can negatively affect its measurements and results, hence it is preferable putting it on a bedside table, lying still, instead of putting it on the mattress. As a consequence, the sonar is often at least 0.5 m away from the patient. This problem also happens with similar radio frequency technology used to monitor the body movements and breathing.

One of the main factors that influence the accuracy of sleep detection devices is the quality of their sensors. An informative discussion and comparison of sensors’ accuracy appears in Lee & Finkelstein (2015). Another important factor is the software that process the data collected by the sensor. Currently, there are more than 100,000 health apps in the Apple and Google Play app stores (Research 2 Guidance, 2016). Many of these apps focus on sleep, and a large proportion of them implement proprietary sleep detection algorithms. As a consequence, the same device (e.g., a mobile phone with an accelerometer) can produce different results depending on the underlying software that process the data collected.

It is therefore very important to highlight that most of the publicly available sleep apps have not been clinically validated. Most of them are implemented and maintained by independent (non-clinical) programmers and, thus, their clinical use is not recommended. Of course, there have been many studies devoted to validating hardware devices and reporting on their accuracy and precision. Some studies devoted to validating actigraphs are Sivertsen et al. (2006), Paquet, Kawinska & Carrier (2007), Sitnick, Goodlin-Jones & Anders (2008), Montgomery-Downs, Insana & Bond (2012), Marino et al. (2013), Meltzer et al. (2014), De Zambotti, Baker & Colrain (2015), Bhat et al. (2015), Toon et al. (2016) and Meltzer et al. (2016). They all study the correlation between one commercial actigraph and a PSG (the patient wore the actigraph during the PSG). The sensitivity reported is in the interval 86%–98%, and the specificity is in the interval 20%–54%, because commercial actigraphs prioritize sensitivity over specificity. The interested reader is referred to Evenson, Goto & Furberg (2015) and Kolla, Mansukhani & Mansukhani (2016) where systematic reviews of validation studies for sleep detection hardware devices can be found. Other works study the precision of actigraphy with specific populations (children, adults, old women, mentally disordered, etc.) (Blackwell et al., 2008; Martin & Hakim, 2011; Marino et al., 2013; Baandrup & Jennum, 2015; Min et al., 2014; De Zambotti et al., 2015; Meltzer et al., 2016).

The review of the state of the technology together with the review of the validation studies advise against using contactless devices in the clinical study of sleep. Their low precision renders them far from being a reliable method. This does not mean that they are useless. They are good sleep indicators, and a good resource for patients to monitor and be aware of their own sleep quality. But their use as a definitive diagnostic tool is to be discouraged. In the case of contact devices, their precision is acceptable for many populations. In general, they should be used as an indicator but not as a definitive diagnostic tool, because several studies report that their sensitivity can fall down to 86% and their specificity to 20%. However, these numbers are on the increase because the advances in technology are continuously improving such devices. This is also observable in the continuous increment of precision reported over time by validation studies.

Conclusions

The first conclusion of this review is that a perfect sleep assessment method does not exist. All methods have advantages and disadvantages, thus, they should be combined and adapted to the specific applicable needs. In terms of accuracy, the PSG is the best method, reporting the most complete and precise information (e.g., differentiating the sleep phases). Nevertheless, PSGs are expensive, exclusive (they require special hardware and medical support), can only be administered once, or for a few days, and they assess sleep in a stressful context (e.g., a hospital with video cameras recording and several machines registering the information provided by electrodes placed in the patient’s body).

For these reasons, sleep diaries and questionnaires are often used to complement the PSG. They provide information that is gathered over medium to long periods of time, including information about sleep habits. Because they are mostly subjective, they have been erroneously considered as unreliable. But, in contrast, several studies (see Table 4) have proven that their sensitivity is often above 90%, and between 73% and 97.7% in all the discussed studies. Specificity ranges in the interval 50%–96%. In the specific case of electronic diaries, the studies demonstrate that they produce the same results as their paper counterparts, but also that they provide functional advantages: automatic data processing, metadata such as information about when the patient filled in the diary, alerts, etc. Sleep questionnaires and diaries have been classified in Tables 2 and 5, respectively.

The literature review shows that the accuracy of hardware devices is superior to that of questionnaires. This superiority, however, must be considered only in terms of precision, but not in terms of diagnostic usefulness. The information provided by questionnaires regarding self-perception of sleep quality is essential and cannot be replaced by hardware measures. The sensitivity of hardware devices is 88–98%, while their specificity is 20–52%. The adherence of hardware devices is also superior, because they require less effort from the patient (e.g., actigraphs are mostly automatic). The accuracy and reliability of hardware devices have been continuously increased with the advances of the technology. The continuous improvement of sensors and the appearance of new technologies (good examples are the imminent use of infrared thermometers and lidars) clearly improve the sleep detection devices.

We have presented a taxonomy of sleep methods that comprises all methods presented. This taxonomy classifies hardware devices into contact and contactless devices, because their functionality, accuracy, and reliability are different. In both cases, many studies reach the similar conclusion that current sleep trackers are useful tools to assess sleep and have been used successfully in many sleep studies. In particular, the studies that have evaluated concrete actigraphs with their respective software/app conclude that actigraphy is a reasonably reliable method to detect sleep with an average sensibility higher than 90%. Nevertheless, the studies also report that the results obtained are particularly influenced by the patient that wears the actigraphy, which can produce bad results in many cases. Therefore, the use of actigraphy as a diagnostic tool should be complemented and contrasted with other methods to produce a more definitive diagnostic.

Finally, sleep apps are another important tool to assess sleep, especially in smartphones, where they are becoming very common nowadays. We have shown the most significant apps with regard to their number of reviews, and to their overall mark given by users. It is important to highlight that, contrasted with hardware, software apps are often implemented by independent (non-clinical) developers, and they do not pass any quality test. Therefore, they must be validated, at least before they are applied in clinically. The few scientific validation studies that have compared smartphone apps against PSG report that they are still not accurate enough to be used as clinical tools.

The authors would like to thank Roger Nolan for reviewing a preliminary version of this article. His help to improve our work is greatly appreciated.

Abbreviations List

ARI arousal index

ARP arouse period

ASQ Athens sleep questionnaire

AWI awakening index

AWP awake period

BQ Berlin questionnaire

BIS Bergen insomnia scale

BIPAP bi level positive airway pressure

CPAP continuous positive airway pressure

CSD consensus sleep diary

CTT CPAP titration test

EEG electroencephalogram

EKG electrocardiogram

EMG electromyogram

EOG electrooculogram

ESS Epworth sleepiness scale

FS final sleep

FOB final out bed time

FOSQ functional outcomes of sleep questionnaire

GASP graduated apnea screening protocol

GERD gastroesophageal reflux disease

GSH get self help sleep diary

HST home sleep test

IBT in bed time

IIB initial in bed time

ISI insomnia severity index

ISL initial sleep latency

LIDAR light detection and ranging

LN lights on time

LSRC Loughborough sleep research center

LT lights out time

MAL mean awakening length

MSL mean sleep latency

MSLT multiple sleep latency test

MWT maintenance of wakefulness test

NHLBI national heart, lung, and blood institute

NPS MedicineWise sleep diary

NSF national sleep foundation

OSA50 obesity, snoring, apneas, aged over 50

OSAHS sleep apnea hypopnea syndrome

OSQ Oviedo sleep questionnaire

PLMS periodic limb movements during sleep

PSD Pittsburgh sleep diary

PSG polysomnogram

PSQI Pittsburgh sleep quality index

QoL quality of life index

RADAR radio detection and ranging

ROC receiver operating characteristic

SACS sleep apnea clinical score

SAQLI Calgary sleep apnea quality of life index

SBQ STOP-BANG questionnaire

SDB sleep-disordered breathing

SDQ sleep disorders questionnaire

SE sleep efficiency

SEMSA self-efficacy measure for sleep apnea

SFV simple four variables

SI sleep interval

SL sleep latency

SO sleep onset

SONAR sound navigation and ranging

SP sleep period

SQ STOP questionnaire

TOB time out of bed

TRT total recording time

TST total sleep time

TWT total wake time

WASO wake after sleep onset

Additional Information and Declarations

Competing Interests

Author Contributions

Data Availability

The authors declare there are no competing interests.

Vanessa Ibáñez, Josep Silva and Omar Cauli conceived and designed the experiments, performed the experiments, analyzed the data, contributed reagents/materials/analysis tools, prepared figures and/or tables, authored or reviewed drafts of the paper, approved the final draft.

The following information was supplied regarding data availability:

This article is a survey and did not use raw data.

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
