# Peer review of "A survey on sleep assessment methods"

_PeerJ, doi:10.7717/peerj.4849_

## Round 0.1 · original submission · Major Revisions

Dear Authors,

Two reviewers have requested major changes to your manuscript.

Thank you.

Reviewer 1 ·

Basic reporting

No comment.

Experimental design

From my point of view, authors could follow the Preferred Reporting Items
for Systematic reviews and Meta-Analyses (PRISMA)
guidelines (Liberati et al., 2009).

Validity of the findings

As a general advice, authors could add some tables summarizing the findings of previous studies on each method reviewed in their work. This could give a clearer idea to the reader on the validity of each method.

Additional comments

Manuscript ID: 23959v1

Journal: PeerJ

Title: A survey on sleep assessment methods


Abstract

1) Lines 44-45. Authors wrote that “Objective methods such as actigraphy present a sensibility higher than 90%.” Here they could also state that specificity of actigraphy is lower compared to sensibility, being one of the limitation of such technology.

2) Line 46. Authors wrote “such as the patient’s perception of his sleep”. Please add “her” before or after “his”.


Introduction

1) Lines 56-57. Authors wrote that “such, most health care systems invest a lot of money in the creation of sleep centres and sleep units in hospitals”. Is there any reference supporting such sentence?

2) Lines 67-69. Authors wrote that “The main motivation of this survey is to produce a comprehensive and unbiased literature review from which we can extract a complete classification of sleep assessment methods (including new technologies such as mobile apps).” I suggest that authors explicit the main features of examined populations, e.g., children, adolescents, adults, healthy controls, patients (if patients, which kind of disorder).

3) Line 98. I suggest that authors follow the Preferred Reporting Items for Systematic reviews and Meta-Analyses (PRISMA) guidelines (Liberati et al., 2009).

4) Line 98. Authors could show the flow-chart of the bibliographic search for each database, according to the PRISMA guidelines.


Classification of sleep detection methods

1) Line 156. Authors should specify here that PSG is the gold standard for the sleep assessment.

2) Lines 180-182. The same sentence also applies to sleep diaries.


Medical Assistance Methods

1) As a general advice, authors could add some tables summarizing the findings of previous studies on each method reviewed in their work. This could give a clearer idea to the reader on the validity of each method.


Self-assessment methods

1) Line 316. Authors should add “her” before or after “his”.


Hardware devices

1) Line 406. Zeo acquires a single electrical signal composed by brain activity, muscle tone and eye movement. Therefore, Zeo cannot be considered an actigraph.


Tables

1) Table 1. The definition of total time out of bed should be the interval between FOB and IIB.


2) Table 2. Authors could revise the works on the Mini Sleep Questionnaire (e.g., Falavigna et al., 2011; Natale et al., 2014), a self-assessment tool of both sleep and wake originally proposed by the research group coordinated by Professor Peretz Lavie (Zoomer et al., 1985), aiming to assess if it fulfils their inclusion criteria.


Figures

1) Figure 1. From my point of view, actigraphy should be considered as a technology requiring medical assistance, because actigraph is a medical device.

2) Figure 1. In the legend, authors should specify the meaning of acronyms reported in such figure.

Reviewer 2 ·

Basic reporting

The review is interesting, informative and coherent with the aims of the journal, but several major revisions are needed.

Major points
- The main problem concerns the originality of the paper. Authors recently published a paper on the journal “Sleep Medicine” in which they reviewed the current methods to assess sleep based on questionnaires and diaries (cit. Ibanez V., Silva J., Cauli O. A survey on sleep questionnaires and diaries. Sleep Medicine, 2018; 42:90-96). Albeit the present paper is a more extensive and general survey on sleep assessment methods, I see an overlap with the “Sleep Medicine” review. Furthermore, such paper is not even cited. Clearly, a partial overlap is unavoidable, but I think that it should be reduced. The “Sleep Medicine” paper should be cited, and the originality of the dissertation on questionnaires and diaries should be improved. Moreover, Table 2, 3 and 4 contain information that are largely reported in the (very similar) tables in Ibanez et al., 2018, so they should be eliminated or strongly differentiated by the old ones (e.g. improving their informative value).
- I suggest a complete revision of the English across the text (it is often wrong, unclear or repetitive).

Abstract
- The “results” section of the abstract actually does not report results, but methods.

Medical Assistance methods
- for each described method a) the principal studies should be cited (at present, I see references only for PSG, and they are not exhaustive); b) the main sleep disorders/problems that can be assessed should be reported.
- Lines 272-273: describe what sensors are used and how the device detects sleep stages.

Hardware devices:
- For both contactless and contact devices, a table reporting the devices specifics (the name of the devices, ranking/review, variables measured by the devices) could be useful.

Contact hardware devices:
- Lines 406-407: It seems that Zeo Personal Sleep Manager is presented as an actigraph, while it detects simplified EEG measures.

A critical discussion about accuracy and validation
- lines 429-430: authors should cite the studies at the basis of their ranking

Figures
- Fig. 2: the figure should be improved: a) it should be referred to a diagnostic classification system; b) for each disorder, authors should report at least one relevant study that use PSG for the assessment of that disorder. Moreover, the present figure is incomplete (e.g. insomnia is missing) and peculiar (medical-psychiatric sleep disorders are reported as parasomnias?).



Minor points
- line 205: the brief summary for each method is redundant, since the cited methods are described later. I suggest the removal of the summary and the integration of the information that it contains in the main description of the methods.
- line 305: “polysomnogram” should be abbreviated
- Fig. 1: abbreviations should be reported in the figure legend

Experimental design

no comment

Validity of the findings

no comment

---

## Round 0.2 · accepted · Accept

Congratulations. Based on the positive assessment of the re-reviewers, I am happy to Accept your manuscript.

Reviewer 1 ·

Basic reporting

Please refer to the general comments for the author.

Experimental design

Please refer to the general comments for the author.

Validity of the findings

Please refer to the general comments for the author.

Additional comments

Manuscript ID: 23959v2

Journal: PeerJ

Title: A survey on sleep assessment methods



The authors have adequately addressed my previous concerns. I do not have any further suggestions.

Reviewer 2 ·

Basic reporting

The authors made all the required changes.

Experimental design

No comment.

Validity of the findings

No comment.